# Risk Management of COVID-19 in the Residential Educational Setting: Lessons Learned and Implications for Moving Forward

**DOI:** 10.3390/ijerph18189743

**Published:** 2021-09-16

**Authors:** Anna L. Cass, Meghan M. Slining, Connie Carson, Jason Cassidy, M. Carmela Epright, Ann E. Gilchrist, Kenneth Peterson, John F. Wheeler, Natalie S. The

**Affiliations:** 1Department of Health Sciences, Furman University, Greenville, SC 29613, USA; anna.cass@furman.edu (A.L.C.); meghan.slining@furman.edu (M.M.S.); 2Division of Student Life, Furman University, Greenville, SC 29613, USA; connie.carson@furman.edu (C.C.); jason.cassidy@furman.edu (J.C.); 3Department of Philosophy, Furman University, Greenville, SC 29613, USA; carmela.epright@furman.edu; 4Furman Earle Student Health Center, Prisma Health, Greenville, SC 29613, USA; ann.knowles@prismahealth.org; 5Office of Academic Affairs, Department of Economics, Furman University, Greenville, SC 29613, USA; ken.peterson@furman.edu; 6Office of Academic Affairs, Department of Chemistry, Furman University, Greenville, SC 29613, USA; john.wheeler@furman.edu

**Keywords:** COVID-19, prevention and mitigation, university student health

## Abstract

With limited COVID-19-guidelines for institutions of higher education (IHEs), colleges and universities began the 2020–2021 academic year with varying approaches. We present a comprehensive COVID-19 prevention and mitigation approach at a residential university during the 2020–2021 academic year, along with campus SARS-CoV-2 transmission during this time. Risk management of COVID-19 was facilitated through (1) a layered approach of primary, secondary, and tertiary prevention measures; (2) a robust committee structure leveraging institutional public health expertise; (3) partnerships with external health entities; and (4) an operations system providing both structure and flexibility to adapt to changes in disease activity, scientific evidence, and public health guidelines. These efforts collectively allowed the university to mitigate SARS-CoV-2 transmission on campus and complete the academic year offering in-person learning on a residential campus. We identified 36 cases of COVID-19 among the 2037 in-person learners during the fall semester, 125 cases in the inter-semester break, and 169 cases among 2095 in-person learners during the spring semester. SARS-CoV-2 infection during the academic year was associated with gender (*p* = 0.04), race/ethnicity (*p* = 0.01), and sorority/fraternity membership (*p* < 0.01). Infection was not associated with undergraduate vs. graduate student status, Division I athlete status, or housing type (all *p* > 0.05). A multi-faceted public health approach was critical for reducing the impact of COVID-19 while carrying out the university’s educational mission.

## 1. Introduction

After rapid campus closures and near-universal pivots to remote instruction in Spring 2020 due to the COVID-19 pandemic, universities were forced to plan for the 2020–2021 academic year in the midst of emerging information and conflicting guidance. Additionally, enrollment declines of 5–20% were predicted for the upcoming fall semester, with greater economic impacts anticipated for private institutions [1]. Colleges and universities faced a prospective student market in flux, with student surveys showing high proportions delaying college decisions, considering “gap” years, and communicating expectations of decreased tuition and fees if institutions were to proceed with remote instruction [2]. Emphasis on in-person and experiential learning presented further challenges at liberal arts and sciences universities. Universities responded to the unprecedented and unpredictable situation with varied responses [3]. To date, most published reports have focused on the experiences of larger and primarily non-residential universities [4,5,6].

We describe the process and partnerships involved in implementing public health measures in response to COVID-19 at a small, residential liberal arts and sciences institution during the 2020–2021 academic year. We present outcomes, along with challenges and successes, identifying aspects of the response structure that could provide a model to ongoing and future public health efforts in university settings.

## 2. Materials and Methods

### Setting and Population

The university is located in Greenville County, South Carolina, a geographic area that implemented minimal COVID-19 mitigation mandates. In summer 2020, Greenville County experienced its first wave of COVID-19 cases with two-week incidence rates peaking at 500 cases per 100,000 individuals (13 July 2020–27 July 2020) and percent positivity tests greater than 15% [7]. During August and September, COVID-19 incidence declined with two-week incidence ranging from 150–200 cases per 100,000. The county entered its second wave beginning in October (234.7 cases per 100,000 from 1 October 2020 to 14 October 2020) and peaked at 1767 cases per 100,000 between 30 December 2020 and 12 January 2021 with percent positive tests in excess of 30%. COVID-19 cases receded during spring 2021.

In the 2020–2021 academic year, the university took a hybrid education approach, relaxing on-campus living requirements and allowing options for remote learning. In this dynamic student population, the majority of the 2310 enrolled students (fall semester, *n* = 2095; spring semester, *n* = 2109) elected to participate in on-campus learning and/or housing. Students were majority undergraduates (98.5%) and female (60.6%) and resided on campus for at least one of the semesters (91.3%). Of reported races and ethnicities representing ≥5% of the student population, 78.1% were non-Hispanic white, 6.8% were non-Hispanic Black, and 5.0% were Hispanic. Fourteen percent of students were involved in Division I athletics, 46.0% were affiliated with a sorority or fraternity, and the mean age was 20.4 years (SD = 2.5). This report focuses on those students living or learning on campus. Of the 30 weeks of in-person learning, 23 weeks were in a period in which the county was defined as having high transmission, while the remaining 7 weeks had substantial transmission.

## 3. Methods

In preparation for the academic year, the university established several core principles to guide their health and safety efforts and support their goal of safely teaching students in an in-person setting [8]. Given the changing circumstances and emerging knowledge about COVID-19, university administration recognized the need for flexibility and creativity alongside a formal, systematic process to facilitate operations and communication between on-campus teams. The resulting processes and actions are outlined below.

### 3.1. Public Health Planning and Response Structure and Collaborations

#### 3.1.1. Internal University Structure

Leveraging the expertise of key partners across university divisions was critical for reducing the impact of COVID-19 while carrying out the university’s educational mission. In March 2020, the university’s Senior Administrative Team (President and Vice Presidents) relied on guidance from existing committee structures, actions at other universities, and consultations with public health faculty to pivot to remote learning following an extended spring break and to bring students home from study away programs. In early summer 2020, the university administration developed a tiered committee structure to assess and develop public health responses and operationalize COVID-19 mitigation measures for the coming academic year. The primary committees included the: (1) COVID Response Steering Committee; (2) Public Health and Safety Advisory Group (PHSAG); (3) COVID Operations Teams; and (4) Dashboard Data Group (Figure 1). More than twenty additional subcommittees were created during the summer to develop and operationalize changes (Table 1).

Committees and subcommittee membership represented diverse expertise and responsibilities across the university, with membership overlap across committees to facilitate consistency. Some subcommittees were eliminated or folded into existing committees over time as implementation of their work was completed or inefficiencies were identified. The streamlined committee structure was maintained throughout the spring semester with the addition of committees as needed, such as a vaccine committee. Recommendations from these committees advanced to the SAT, which worked with a Board of Trustees COVID-19 Oversight Task Force for final decision making.

Throughout the process, university administration participated in weekly calls to collaborate with consortium institutions and other institutions of higher education within the state.

#### 3.1.2. External Collaborations

The university collaborated with four external entities to facilitate campus operations. An existing partnership with a healthcare system provided early clinical guidance and diagnostic testing capabilities through the on-campus student health center. A private local laboratory performed reverse transcription–polymerase chain reaction (RT-PCR) tests on campus for asymptomatic testing, first for university athletics and later for university-wide surveillance. This collaboration enabled routine testing of large numbers with 24 h turnaround time for results. South Carolina Department of Health and Environmental Control (SCDHEC) provided contact tracing guidance prior to the reopening of campus. At the end of the fall semester, SCDHEC also provided on-campus testing for the university and local community prior to the close of in-person learning in anticipation of Thanksgiving weekend travel. Additionally, university faculty collaborated with content experts at SCDHEC to provide a panel discussion on COVID-19 vaccines to students and employees prior to an on-campus vaccine clinic. This one-day vaccine clinic for students and employees was accomplished in collaboration with another healthcare system.

### 3.2. Implementation of Prevention and Mitigation Strategies

#### 3.2.1. Changes to Academic Calendar

The academic year started one week earlier than usual, on 18 August 2020, to allow in-person instruction to end prior to Thanksgiving (20 November 2020), with end-of-semester coursework and exams conducted online to minimize travel between holidays. Additionally, a staggered start to the fall semester allowed for reduced population density on campus and facilitated a flexible operational response for the newly implemented policies and procedures. First- and fourth-year students, those least likely to share housing or academic spaces, returned first to campus for in-person learning; second- and third-year students returned three weeks later (7 September 2020).

Due to a national spike in infection rates in late December and early January, the start of the spring semester was delayed one week to 19 January 2021, with virtual-only classes the first week. Single break days were scheduled throughout the spring semester, replacing the typical weeklong Spring Break, in order to discourage widespread travel.

#### 3.2.2. Primary Prevention

A layered prevention approach was implemented, following Centers for Disease Control and Prevention (CDC) and SCDHEC guidelines available at the time. Primary prevention strategies included required masking and physical distancing of six feet with minimal exceptions (e.g., roommates in their private residences when guests were not present). Students were encouraged to reduce the number of social contacts. Class capacities were reduced. Indoor dining was limited to students, and table capacity was reduced. In-person engaged learning experiences (e.g., internships, study away, service learning) and large social gatherings were not permitted. Enhanced cleaning and ventilation were implemented. The campus was closed to visitors, and the on-campus fitness center was limited to students enrolled in a related academic class (by early October all students were permitted access).

#### 3.2.3. Secondary Prevention

Students and employees completed daily self-screening through a survey connected to an existing campus app.

At the start of the fall semester, CDC and SCDHEC guidelines did not yet recommend widespread surveillance testing, given the limited availability of tests and potential impact on testing laboratories [9,10]; thus, this approach was not implemented initially. Following an outbreak described further in Fall Semester Events, surveillance testing was implemented and continued throughout the spring semester. A random sample of students was selected weekly for testing, with flexibility to allow for additional risk-based sampling in response to observed patterns among cases. Sampling was set at 20% of on-campus students for most of the fall, with the proportion varying dependent upon community and on-campus disease activity. A testing site was established on campus, and testing was conducted two days a week.

#### 3.2.4. Tertiary Prevention

Student health services provided in-person and remote health evaluations, diagnostic testing, and clinical care through the on-campus student health center.

Approximately 90 housing spaces were designated for quarantine and isolation. Students were asked to make a COVID-19 contingency plan prior to the start of the fall semester and encouraged to plan for potential quarantine or isolation at home when feasible.

Once a student tested positive for SARS-CoV-2, the on-campus student health services contacted the student for clinical care and notified the university employee who served as the university’s contact tracer. The tracer queried the student by phone about close contact (within 6 feet of someone for 15 consecutive minutes or more in one 24 h period) with other individuals at the university in the 48 h prior to receiving a positive SARS-CoV-2 test result. All individuals determined to be close contacts were then notified that they were required to enter quarantine, either off-campus or in designated campus housing. During the fall semester, per CDC guidelines, all students remained in quarantine for 14 days. In the spring semester, CDC guidance allowed individuals in quarantine to test 5–7 days after exposure and to exit quarantine with a negative result.

### 3.3. Fall Semester Events and Public Health Responses

In the first weekend of the fall semester, two unsanctioned fraternity events occurred off campus, resulting in an outbreak of COVID-19 among students. Public health actions included immediate quarantine and testing of all known student attendees, with 34 of 58 students testing positive for SARS-CoV-2. Testing was required for all remaining students (*n* = 1258) later that week. Second- and third-year students were then required to submit a negative test completed within five days prior to arrival on campus.

Recognizing the importance of testing in mitigating the impact of the early-semester outbreak, along with increasing availability of testing and emerging evidence in the literature supporting widespread surveillance testing [11,12], PHSAG recommended to SAT the implementation of routine surveillance testing of the student body. As a result, surveillance testing was conducted in addition to diagnostic testing for the remainder of the academic year, managed by faculty epidemiologists and operational committees.

Reports of student participation in an off-campus Halloween pub crawl resulted in required COVID-19 testing for all third- and fourth-year students and a low-contact period for all students until testing results were available.

### 3.4. Inter-Semester Disease Activity and Public Health Planning

COVID-19 incidence in Greenville County spiked in late December and early January, with the highest two-week incidence to-date of 1756.5 per 100,000 [7]. Additionally, hospital bed utilization became a significant concern, with local hospitals cancelling elective procedures. Due to community disease activity, the spring semester began with additional prevention and mitigation measures in place. In addition to a delayed start and online-only classes the first week, students were required to submit a negative test within five days prior to arrival, and all students were tested in each of the first two weeks of the semester. Students were required to remain in small pods (≤4 individuals), meals were only available to-go, and the fitness center was closed to students except as needed for academic courses.

### 3.5. Development and Implementation of Operational Phases

Utilizing broad public health guidelines for institutions of higher education (IHEs), epidemiologists within PHSAG developed a framework specific to our institutional context to guide campus operations based on on- and off-campus COVID-19 measures [13]. The color-coded operational phases system was tested internally using fall semester data and was approved for use during spring 2021 (Table 2). PHSAG reviewed data weekly to make recommendations for the following week’s operational phase.

Recognizing that full restrictions for a prolonged period were not sustainable, a harm-reduction approach [14] was adopted to balance the social and emotional well-being of young adults with the risks of COVID-19 for students and the broader campus community (including potential higher-risk employees).

After establishing a baseline of <1% weekly campus incidence during the first two weeks of the spring semester amid high county-level incidence, campus operations proceeded in Orange phase. Surveillance testing was subsequently reduced to 50% of students for the following two weeks. As elective procedures at regional hospitals resumed and on-campus disease transmission remained low, the campus moved to Yellow phase by mid-February and decreased surveillance testing to 25% of students, and subsequently to 20%, following testing of 200 students with 0 positive results. With low on-campus infection and improving community levels, the university entered Purple phase from the end of March to the end of the semester. On-campus student gathering restrictions were relaxed, and surveillance testing was reduced to 15% of students; restrictions remained in place for off-campus activities.

### 3.6. Spring Semester Events and Public Health Responses

The weekend before classes ended, at least 39 students attended an unauthorized on-campus gathering at which disease precautions were not followed. Public health actions included quarantine and testing for involved students. At the time of the event, 11 of the 39 student attendees were fully vaccinated, and eight were within 90 days of previous SARS-CoV-2 infection. The remaining students were tested, and no infections were identified associated with this event.

### 3.7. Data and Analysis for COVID-19 Outcomes

University administration created and maintained a student database to support COVID-19-related actions and university functions. We analyzed the database for this report and present descriptive statistics and bivariate analyses of student characteristics and SARS-CoV-2 infection. Chi-squared tests compared SARS-CoV-2 infection across student groups; Fisher’s exact test was used for characteristics with small sample sizes. Risk ratios and 95% confidence intervals (CIs) were calculated to quantify the relative risk of COVID-19 across different student groups.

## 4. Results

### COVID-19 Disease Outcomes 

Considering only the time students were active on campus during the academic year, 13.4% of students experienced a SARS-CoV-2 infection during the academic year with 136 cases of COVID-19 identified among the 2037 in-person learners in the fall semester and 169 cases among the 2095 in-person learners during the spring semester. SARS-CoV-2 infection during the academic year was associated with gender (*p* = 0.04), race/ethnicity (*p* = 0.01), and sorority/fraternity membership (*p* < 0.01). Differences were not observed across undergraduate/graduate student status (*p* = 0.08), Division I athlete status (*p* = 0.2), or housing type (*p* = 0.08) (Table 3). The relative risk of male students compared to female students was 1.25 (95% CI: 1.01, 1.54), of white non-Hispanic students relative to Black non-Hispanic students was 2.41 (95% CI: 1.27, 4.59), and members of a sorority or fraternity compared to those not affiliated with a sorority or fraternity was 1.70 (95% CI: 1.37, 2.11).

Percentage utilization of on-campus quarantine and isolation housing ranged from 0–20% over the academic year. During the fall semester, 18.9% (396/2037) of students were identified as a close contact and placed in quarantine. Twelve of these students became symptomatic during quarantine but tested negative for a SARS-CoV-2. An additional 18 students became symptomatic during quarantine and tested positive for SARS-CoV-2 infection. Eight of eighteen were roommates, significant others, or teammates of the initial individual who tested positive. During the spring semester, 18.4% (386/2095) students were identified as a close contact and placed in quarantine. Seven of these students became symptomatic during quarantine but tested negative for a SARS-CoV-2. An additional 24 students became symptomatic during quarantine and tested positive for SARS-CoV-2 infection. Eleven of eighteen were roommates, significant others, or teammates of the initial individual who tested positive.

Six percent of spring semester students reported infection with SARS-CoV-2 during the inter-semester 6-week break (125/2095). Infection during the winter break was associated with athlete status (8.0% of athletes (29/361), vs. 5.1% of non-athletes (96/1888), *p* = 0.03). No significant associations were observed between other student characteristics and infection among the 5.6% of the in-person student population infected during this period.

Weekly surveillance testing resulted in percent positivity ranging from 0% to 1.3% (Figure 2). In the fall semester, 40% of cases identified on campus were identified through surveillance testing. In the last two months of the spring semester, presentation of COVID-19 cases shifted on campus, with only five positive results from surveillance testing out of 2310 tests conducted. During this same time, the student health center conducted 193 COVID-19-related evaluations, resulting in 108 referrals for testing, of which 51 returned positive for SARS-CoV-2.

## 5. Discussion

### 5.1. Successes

This university developed and implemented a comprehensive risk management approach through a robust committee structure with public health expertise to fulfill the university’s core mission to provide in-person, engaged learning experiences amidst a pandemic. Uninterrupted in-person learning was possible with a layered disease prevention and mitigation approach with flexibility to respond rapidly to disease levels on- and off-campus. While the university experienced COVID-19 cases each semester, mitigation efforts minimized adverse consequences of COVID-19 to the campus population. The number of cases in a 15-week semester at a university with congregate living settings was nearly identical to that of the 5-week break between semesters (when students returned home) and weekly cases remained relatively low.

Central to the ability to safely provide in-person learning were effective internal and external processes and partnerships to address the volume and changing nature of issues. The continual availability of a public health and safety advisory group to make recommendations for campus decision-making needs in light of dynamic data and emerging public health guidelines, combined with representation of senior leadership in the committee structure, facilitated the ability to enact recommendations quickly and ensure timely allocation of resources.

### 5.2. Lessons Learned

Harm reduction provided a helpful framework for considering the potential unintended negative consequences of and tension between different possible actions, recognizing the physical and mental health needs of students.

The formation of the PHSAG provided a structure specifically focused on public health goals, facilitating review of emerging public health guidelines, assessment of potential public health actions, and formulation of targeted recommendations to be assessed by other committees in the context of university operations. A liaison between this advisory group and the steering committee facilitated workflow and communication.

Strengths of the committee structure included working groups of manageable size to maximize productivity, overlap of membership across committees to facilitate continuity of efforts, and the flexibility to respond to emerging needs as required. The breadth of expertise involved in committees—ranging from public health to housing to police—ensured the application and enforcement of disease prevention and mitigation measures in a way that functioned within existing university practices and structures. Student membership in committees provided an important student voice for decision-making.

### 5.3. Challenges

Reopening a university for in-person learning after a swift pivot to remote learning in spring 2020 semester presented significant challenges. Guidance for universities from public health agencies was still evolving and limited at critical times when university planning and operational decisions needed to be made for the Fall 2020 semester.

The reallocation of human resources required to accomplish COVID-19-related decision-making, implementation, and enforcement was significant, and burnout was a concern. Students and employees faced COVID-19 fatigue, requiring the university to adjust key support services.

### 5.4. Limitations

The findings in this report include limitations. The university consists of a dynamic college population, with some students returning home for short times in the middle of the semester or permanently. Disease activity was analyzed from a database of students completing each semester of in-person learning and may not represent those who withdrew or switched to remote learning mid-semester. The university’s National Collegiate Athletic Association Division I athletics program followed a separate protocol for student athletes not described in this report. The results presented in this case study may not be generalizable to large or non-residential universities.

## 6. Conclusions

Multiple layers of prevention mitigated the COVID-19 impact at a small university and allowed for in-person learning without interruption. As universities navigate operational decisions for the 2021–2022 academic year, they are again faced with changes that will necessitate further revision of the strategies developed. Universities are now faced with decisions regarding potential vaccine requirements, which would provide an additional layer of infection prevention to a campus setting but may be hindered by state legislation and cultural perceptions. Vaccination rates of students and employees must be factored into restrictions for the summer and leading into the next academic year. High vaccine coverage on university campuses will be an important element of robust risk management in educational settings, particularly in light of emerging and highly transmissible variants.

## Figures and Tables

**Figure 1 ijerph-18-09743-f001:**
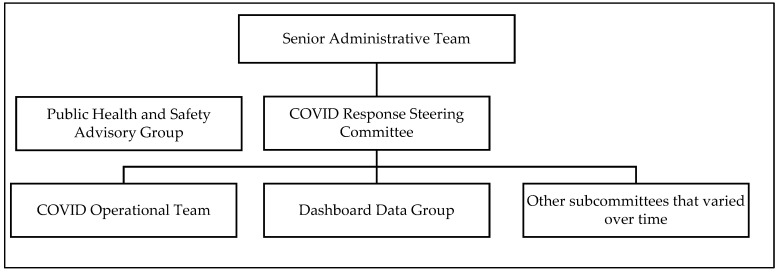
University COVID-19 Committee Structure—South Carolina, 2020–2021 Academic Year.

**Figure 2 ijerph-18-09743-f002:**
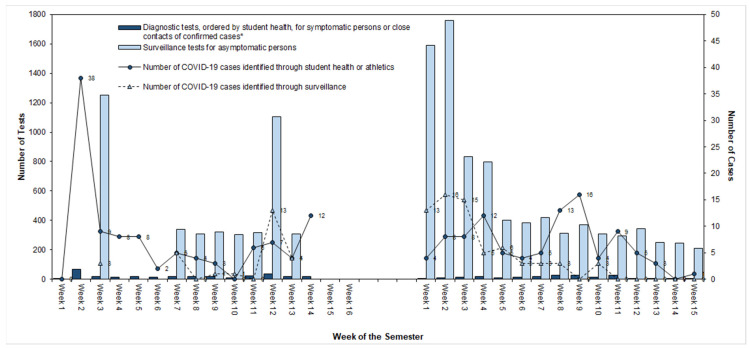
Number of RT-PCR COVID-19 tests performed by testing strategy and number of COVID-19 cases on a university campus, South Carolina, August 2020—April 2021. Abbreviations: COVID-19 = coronavirus 2019; RT-PCR = reverse transcription-polymerase chain reaction. * Tests conducted through university Athletics surveillance program data not shown.

**Table 1 ijerph-18-09743-t001:** University COVID-19 Committee Membership and Responsibilities—South Carolina, 2020–2021 Academic Year.

**COVID Response Steering Committee (CRSC)**	
** *Members* **	** *Responsibilities* **
Vice President for Academic Affairs and Provost (co-chair) Vice President for Student Life (co-chair) Dean of Students (lead/facilitator) Vice President for University Communications Assistant Vice President for University Communications Associate Provost for Engaged Learning Associate Provost for Integrative Science Associate Vice President for Enrollment Services Chief of Police Associate Athletics Director/Senior Women’s Administrator	Oversee COVID-19 planning and implementation of approved recommendations
**Public Health and Safety Advisory Group (PHSAG)**	
** *Members* **	** *Responsibilities* **
Associate Provost for Integrative Science (chair) Chief of Police University Student Health Medical Director Associate Athletics Director/Director of Sports Medicine Biomedical Ethicist Epidemiologists (3) University Risk Manager	Make recommendations to CRSC and provide guidance based on emergent research and guidelines Review protocols and proposals developed by subcommittees and university entities Develop surveillance program Review community and on-campus data to recommend university operational phases
**COVID Operational Team**	
** *Members* **	** *Responsibilities* **
**Student Life** Dean of Students (chair) Assistant Dean of Students Director of the Center for Inclusive Communities Assistant Vice President for Student Development Director of Campus Recreation Director of Office of Student Involvement and Inclusion Director of Housing & Residence Life Associate Director, Office of Student Involvement and Inclusion Associate Director of Administration and Finance University Student Health Medical Director **Academic Affairs** Associate Provost for Integrative Science Associate Provost for Engaged Learning Associate Academic Dean Assistant Academic Dean **Athletics** Associate Athletics Director/Senior Women’s Administrator Associate Athletics Director/Director of Sports Medicine Associate Athletics Director/Facilities & Game Operations Assistant Athletics Director/Associate Director of Sports Medicine **Finance and Administration** Associate Vice President for Human Resources Director of Facilities Operations Director of Risk Management Director of Auxiliary Services Chief of Police **Other Divisions** Associate Vice President for Enrollment Services Assistant Vice President for Strategic Communications Executive Director Alumni and Parent Engagement Bon Appetit Resident District Manager	Operationalize COVID-19 protocols on academic instruction, facilities, student and residence life, technology, and human resources Oversee engagement with campus community and public
**Dashboard Data Group**	
** *Members* **	** *Responsibilities* **
Dean of Students Chief of Police University Student Health Medical Director Healthcare Administrator for Athletics University Risk Manager Director of Housing & Residence Life	Collect and review daily data from testing and surveillance Update quarantine and isolation usage Communicate with contact tracing
**Subcommittees and Workgroups**	
Strategic Academic Redesign Team Student Advisory Group Academic and Curricular Subcommittee Accessibility and Accommodations Subcommittee Athletic and Sporting Events Subcommittee Buildings and Building Management Subcommittee Communications Subcommittee Dining Subcommittee Education and Training Subcommittee Engaged Learning Subcommittee External Groups, Alumni, Parents, and Visitors Subcommittee	Fall Retention Subcommittee Housing and Residence Life Subcommittee Orientation and Advising Subcommittee Phased Return to Campus Subcommittee Physical Distancing and Safety Measures Enforcement Subcommittee Prevention, Screening, Testing, and Contact Tracing Subcommittee Student Involvement, Programs, and Activities Subcommittee Winter Break Housing Subcommittee Vendors and Contractors Subcommittee

**Table 2 ijerph-18-09743-t002:** COVID-19 measures used to determine operational phases and campus activities in operational phases in Spring 2021 at a university—South Carolina, January–April 2021.

Measures Informing Campus Operations Phases	
**Campus Measures**	Total number of students with a positive test in the last 7 days
	Percent positivity from surveillance tests
	Percent of campus in quarantine and/or isolation
	Percent of on-campus quarantine and isolation spaces utilized
**Community Measures ***	Upstate SC acute hospital bed occupancy when elective procedures are cancelled
	Greenville County two-week cumulative incidence
	Greenville County two-week percent positivity
	Greenville County two-week incidence trend
**Campus Operations Phases**	
**Green:** **Normal Operations**	COVID-19 indicators in the campus and local community are favorable. Vaccines are available and have been successfully distributed both on campus and in the community. Good public health practices remain in place.
**Purple:** **Basic Precautions**	COVID-19 indicators in the local and/or campus community are at low levels and are trending favorably. Classes are conducted in Flex mode, and campus buildings may be approved to operate under normal hours. Indoor dining is available with a to-go option for students. Face masks and physical distancing are in place, and symptomatic testing is provided for students on-campus. With approval, university-sponsored off-campus activities are permitted, and on-campus activities and community events with more than 30 individuals may be held. Visitors to campus are permitted with limitations. On-campus work for most employees is permitted.
**Yellow:** **Enhanced Precautions**	COVID-19 indicators in the local and/or campus community are at a moderate or elevated level. Classes are conducted in Flex mode. Academic buildings including the fitness center are open with restricted hours and at a reduced capacity. Indoor dining available with a to-go option for students. On-campus activities of up to 30 individuals may be permitted. Face masks, physical distancing, and enhanced testing are in place. University-sponsored off-campus activities may be permitted on a case-by-case basis. Employees participate in approved remote work options.
**Orange:** **High Precautions**	COVID-19 indicators in the local and/or campus community are elevated or trending unfavorably, with potential impact on local healthcare systems. Classes may shift to remote learning. Dining options are to-go only, the fitness center is closed, and employees approved to work remotely are encouraged to remain off-campus. Students are asked to reduce contacts as much as possible. Except for normal class meetings, organized campus activities are limited to 10 individuals, and off-campus activities are suspended or remote. Face masks, physical distancing, and enhanced testing are in place, and students are strongly encouraged to remain on-campus. The campus is closed to visitors, and university-sponsored off-campus activities are suspended unless approved, on a case-by-case basis.
**Red:** **Full Precautions**	COVID-19 indicators in the local and/or campus community suggest widespread transmission with an impact on local healthcare systems. Only essential activities are permitted, which should occur in a virtual setting whenever possible. Campus and building access is limited to essential functions and personnel. Students are required to limit their contacts to roommates, suitemates, or apartment-mates. Dining service is limited to pre-packaged to-go items only. The fitness center is closed, and all employees who are approved to work remotely are required to remain off-campus. Face masks, physical distancing, and enhanced testing measures are in place. The campus is closed to visitors, and both on and off-campus activities are suspended.

* Community measures were reported weekly by South Carolina Department of Health and Environmental Control.

**Table 3 ijerph-18-09743-t003:** Characteristics associated with confirmed COVID-19 cases during the academic year at a university—South Carolina, August 2020–April 2021.

Student Activity or Characteristic	% Students Infected (No. of Cases/No. of Students)	*p*-Value ^†^
**Gender**		0.04
Female	12.2 (167/1364)	
Male	15.3 (135/885)	
**Race ***		0.01
Black, non-Hispanic	6.0 (9/150)	
Hispanic	9.1 (10/110)	
White, non-Hispanic	14.5 (256/1769)	
Other	12.3 (27/220)	
**Housing**		0.08
Dorm (community bath)	14.6 (107/731)	
Dorm (suite bath)	12.6 (38/302)	
On-campus apt	14.4 (131/909)	
Off-campus apt/house	8.0 (12/150)	
Commuter	8.9 (14/157)	
**Sorority or Fraternity Membership**		<0.01
Members	17.2 (180/1044)	
Not Members	10.1 (122/1205)	
**Education Level**		0.08
Graduate	2.9 ^‡^	
Undergraduate	13.6 ^‡^	
**Division I Athletes**		0.20
Athlete	15.5 (56/361)	
Non-athlete	13.0 (246/1888)	

* Categories representing less than 5% of the student population were collapsed in order to protect student confidentiality. The “Other” category includes American Indian or Alaska Native, Asian, Hawaiian or Pacific Islander, Multi-racial, International students, and Unreported. ^†^ The chi-squared test was used to assess the difference in percentage infected between student groups. Fisher’s exact test was used for categories with cell size <5. ^‡^ Frequencies were suppressed due to small numbers to protect student confidentiality. Abbreviations: COVID-19 = coronavirus disease 19.

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
