# Peer review of "Risk Management of COVID-19 in the Residential Educational Setting: Lessons Learned and Implications for Moving Forward"

_ijerph, 2021, doi:10.3390/ijerph18189743_

Round 1
Reviewer 1 Report
Dear Authors, your manuscript is interesting and easy to read. Thus I only have minor comments.
line 90, in figure 1 : it is not very clear how the PHSAG connects to the other structures. It is the case in the text but not in the figure.
line 264, in figure 2: I would recommend adding the semesters starting and ending dates. Or at least in the "changes to academic calendar" section.
With kind regards
Author Response
Dear Authors, your manuscript is interesting and easy to read. Thus I only have minor comments.
Thank you for your kind comments.
line 90, in figure 1 : it is not very clear how the PHSAG connects to the other structures. It is the case in the text but not in the figure.
We have provided an updated figure that includes a dotted line to the Covid Response Steering Committee. Given our expertise, our committee provided additional public health expertise and guidance.
line 264, in figure 2: I would recommend adding the semesters starting and ending dates. Or at least in the "changes to academic calendar" section.
We appreciate the reviewer’s helpful suggestion for clarity and included the dates in the academic calendar section.
Reviewer 2 Report
I am not sure if this study is sufficiently original and provides any additional new knowledge to the field.
Is this about the risk management as “treatment” and examining its effect? If so, the timeline is not clear to me as to when this risk management procedure/protocols got implemented and how the different (colored) phases are supposed to be analyzed. It is also unclear to me what the analysis including the p-values is showing – there is no reference (baseline) group to compare it to, and again what time points the analysis is looking at and comparing to.
“The number of cases in a 15-week semester at a university with congregate living settings was nearly identical to that of the 5-week break between semesters (when students returned home) and weekly cases remained relatively low.” --> is this “relatively low” number low enough? I do not think that these numbers (of cases) are low.
Reviewer 3 Report
The paper addresses a very important public health issue, i.e. how to manage university activities during the COVID-19 pandemic. Some issues deserve more attention, in order the manuscript could be published.
- More details are needed on the university in which the experience was done, in order to increase the generalizability of the finfings.
- The authors talk about primary, secondary and tertiary preventions, but no details on contact tracing activities were provided, Particularly, no info are given on how many secondary cases occurred in the universirity. This info is crucial to understand if the implemeneted measures were effective or not.
- No info are given on the epidemiological trend of the community outside the university. The epidemiological trend of COVID-19 infections should be analyzed together with the context.
Round 2
Reviewer 2 Report
Thank you for addressing my comments. I think at this point that the writing can be tightened to be shorter and in terms of grammar.
Author Response
We appreciate that we have fully addressed the reviewer’s initial comments. We have made editorial and grammatical edits to the newest, revised manuscript.
Reviewer 3 Report
All suggestions have been assessed properly. In my opinion the paper could be published.
Author Response
We appreciate the reviewer’s support.